# Olive Leaves and Hibiscus Flowers Extracts-Based Preparation Protect Brain from Oxidative Stress-Induced Injury

**DOI:** 10.3390/antiox9090806

**Published:** 2020-09-01

**Authors:** Elda Chiaino, Matteo Micucci, Sandro Cosconati, Ettore Novellino, Roberta Budriesi, Alberto Chiarini, Maria Frosini

**Affiliations:** 1Dipartimento di Scienze della Vita, Università di Siena, Via Aldo Moro, 2, 53100 Siena, Italy; chiaino@student.unisi.it; 2Dipartimento di Farmacia e Biotecnologie, Alma Mater Studiorum, Università di Bologna, Via Belmeloro 6, 40126 Bologna, Italy; matteo.micucci2@unibo.it (M.M.); roberta.budriesi@unibo.it (R.B.); alberto.chiarini@unibo.it (A.C.); 3Dipartimento di Scienze e Tecnologie Ambientali Biologiche e Farmaceutiche, Università della Campania Luigi Vanvitelli, Via Vivaldi 43, 81100 Caserta, Italy; sandro.cosconati@unicampania.it; 4Dipartimento di Farmacia, Università degli Studi di Napoli “Federico II”, Via Domenico Montesano 49, 81131 Napoli, Italy; ettore.novellino@unina.it

**Keywords:** *Hybiscus sabdariffa* (L.), *Olea europea* (L.), neuroprotection, oxidative stress, excitotoxicity, natural compounds, neurodegenerative diseases

## Abstract

Oxidative stress (OS) arising from tissue redox imbalance, critically contributes to the development of neurodegenerative disorders. Thus, natural compounds, owing to their antioxidant properties, have promising therapeutic potential. *Pres phytum* (PRES) is a nutraceutical product composed of leaves- and flowers-extracts of *Olea europaea* L. and *Hibiscus sabdariffa* L., respectively, the composition of which has been characterized by HPLC coupled to a UV-Vis and QqQ-Ms detector. As PRES possess antioxidant, antiapoptotic and anti-inflammatory properties, the aim of this study was to assess its neuroprotective effects in human neuroblastoma SH-SY5Y cells and in rat brain slices subjected to OS. PRES (1–50 µg/mL) reverted the decrease in viability as well as the increase in sub-diploid-, DAPI-and annexin V-positive-cells, reduced ROS formation, recovered the mitochondrial potential and caspase-3 and 9 activity changes caused by OS. PRES (50–100 µg/mL) neuroprotective effects occurred also in rat brain slices subjected to H_2_O_2_ challenge. Finally, as the neuroprotective potential of PRES is strictly related to its penetration into the brain and a relatively good pharmacokinetic profile, an in-silico prediction of its components drug-like properties was carried out. The present results suggest the possibility of PRES as a nutraceutical, which could help in preventing neurodegenerative diseases.

## 1. Introduction

In the last decades, a wealth of literature has suggested that despite their clinical heterogeneity neurodegenerative disorders share many common features in the etiology [1]. The accumulation of misfolded proteins is the common hallmark, which occurs alongside neuronal loss in specific brain areas, caused by mitochondrial function impairment, enhancement in endoplasmic reticulum stress as well as neuroinflammation [2]. High levels of oxidative stress (OS) are usually described in the brain of patients suffering from neurodegenerative conditions, and increased formation of reactive oxygen species (ROS) has been proposed as one of the potential crucial steps in their development [3]. Indeed, ROS are not considered as mere “triggering factors”, but rather they are likely to worsen the progression of the disease being produced also during the brain inflammatory response [4,5].

The lack of disease-modifying therapies along with the growing number of cases, makes it critical to search for new approaches for treating different neurodegenerative diseases. The possibility of lowering ROS levels may be a promising strategy to delay or prevent neurodegeneration. In this view, many natural compounds endowed with antioxidant (but not only) properties are currently under clinical investigations (for a review see for example [6,7]).

The pharmaceutical formulation Pres Phytum^®^ (PRES) is a 13:2 (*w*/*w*) mix of the extracts prepared from *Olea europea* L. leaves and *Hybiscus sabdariffa* L. calyces, respectively, marketed as a food supplement (www.nutraceutica.it). This formulation, the composition of which has been fully characterized [8] (see also below) possesses antioxidant and cytoprotective effects [8,9], mainly due to the phenolic compounds found in both plants which includes hydroxytyrosol, oleuropein, flavonoids, chalcones, tannins, and hibiscus acid. Most of them possess neuroprotective properties along with many other positive effects [10,11,12,13]. Therefore, the aim of the present work was to assess the neuroprotective properties of PRES toward OS-mediated injury in human neuroblastoma SH-SY5Y cells. Cell viability, apoptosis markers, ROS formation, as well as changes in the mitochondrial potential were assessed after H_2_O_2_ challenge. Activities of caspase-9 (intrinsic pathway), caspase-8 (extrinsic pathway) and caspase-3 (executioner pathway) were also evaluated. To explore the effects of PRES in a tissue context where the structural and synaptic organization of the original tissue is preserved, thus allowing a better extrapolation in terms of neuroprotection, rat brain slices subjected to OS were also used. Finally, as natural compounds have always faced problems, including limited absorption, rapid metabolism or systemic elimination, as well as inability to penetrate the brain, which might limit their protective activity, in silico physicochemical and pharmacokinetic properties of main PRES components were predicted.

## 2. Materials and Methods

### 2.1. PRES

Pres Phytum^®^ was made up by the 13:2 (*w*/*w*) mixture of the extracts from *Olea europaea* L. leaves (OEE) and *Hibiscus sabdariffa* L. calyces (HSE). Detailed information about the preparation of both extracts, the preparation of the mixture and its composition, assessed by HPLC coupled to a UV-Vis and QqQ-Ms detector, has already been reported [8,9]. Briefly, the main components were (mg/g): in OEE, hydroxyoleuropein (8.96 ± 0.42), elenolic acid glucoside (23.65 ± 0.79), oleuropein (215.1 ± 1.64), oleuropein isomer (51.09 ± 0.35), ligstroside (11.03 ± 0.04), verbascoside (4.67 ± 0.44), rutin (0.69 ± 0.06), luteolin-7-*O*-rutinoside (0.74 ± 0.04), luteolin-7-*O*-glucoside (5.83 ± 0.24), luteolini-4-*O*-glucoside (4.20 ± 0.12); in HSE, hibiscus acid (139.2 ± 0.47 mg/g).

### 2.2. Experiments on Human Neuroblastoma SH-SY5Y Cells

#### 2.2.1. Cell Cultures

Human SH-SY5Y neuroblastoma cells (ECACC Cat# 94030304, passages 6–20), grown in standard conditions, as already reported [14] were sub-cultured at 70–80% confluent, medium changed twice/week and experiments were performed by using cells in the exponential phase of growth.

#### 2.2.2. H_2_O_2_-Induced Injury

SH-SY5Y cells (96-well plates, 1.6 × 10^4^ cells/mL, final volume 200 µl) were treated with 25 µM H_2_O_2_ for 1 h, followed by 24 h with medium: this treatment caused ~50% cell death [14]. H_2_O_2_ was freshly prepared prior to each experiment from a 30% stock solution.

#### 2.2.3. PRES Treatments

PRES stock solution (10 mg/mL in PBS, pH adjusted to 7.3) was prepared immediately before use and carefully filtered (0.45 µm pore size) prior to dilution to the desired final concentration. SH-SY5Y cells were pretreated with PRES for 2 h before adding H_2_O_2_ to the medium and for the following 24 h. To evaluate the effects of PRES per se, cells were incubated with increasing PRES concentrations (0–100 µg/mL) for the corresponding period (2 h of the pretreatment + 24 h).

#### 2.2.4. Cell Viability Assays

MTT assay was used to assess cell viability, which was then reported as the percent of untreated cells (controls) [15]. At the end of the treatments, cell morphology was also carefully checked by a blind expert operator by phase-contrast light microscope and results were expressed using the USP 28 (United States Pharmacopeia edition 2005) grade scale [13,16].

#### 2.2.5. Apoptosis Assays

Cell cycle and sub-G0/G1 population analysis was performed by using flow cytometry. SH-SY5Y (2.5 × 10^5^ cells/mL, final volume of 2 mL) treated as detailed before, were trypsinized, collected by centrifugation (1500 rpm, 10 min), and carefully rinsed before fixing them with 70% ethanol. Before flow cytometry analysis, cells were treated with RNase (0.1 mg/mL, 30 min) and then stained with propidium iodide (PI) (20 µg/mL, 30 min. Red fluorescence (DNA) was detected by FACScan flow cytometer (BD Biosciences, San Jose, CA, USA) in the FL2 channel, acquiring 10^4^ cells/sample. Cell Quest software v. 3.0 (BD Biosciences, San Jose, CA, USA) was used to calculate the percentage of apoptotic cell in the sub-G0/G1 peak and in the other cell cycle phases [16,17]

DAPI (4′,6-diamidino-2-phenylindole) staining kit (Life Tecnologies Italia, Monza, Italy) was used for assessing nuclear morphology. Kit’s suggested protocol, with minor changes (i.e., staining with 0.5 µM DAPI for 10 min, followed by two washing in PBS and one in water) was followed [16,17]. For quantification, nuclei with fragmented and condensed DNA were scored as apoptotic cells by a blind, expert operator, who randomly chose 5–6 fields from at least three independent experiments. [18].

The Alexa fluor 488™-Annexin V/PI double staining kit (Life Tecnologies Italia, Monza, Italy) was used according to the manufacturer’s protocol and data acquisition was performed as previously reported [17]. Cells were scored according the following criteria: PI negative, AV negative: viable cells. AV positive, PI negative, early apoptotic cells. AV positive, PI positive, late apoptotic cells. AV negative, PI positive, necrotic cells.

#### 2.2.6. Intracellular ROS Content

The fluorescence probe 2′, 7′-dichlorofluorescein diacetate (DCFH-DA) was used to detect intracellular ROS level at 1 h or 24 h with an experimental protocol already detailed [13,14]. Intracellular DCF generated by H_2_O_2_-treatment was taken as 100% [13,14].

#### 2.2.7. Mitochondrial Electrochemical Potential Gradient (Δψ)

Mitochondria Staining Kit (Sigma Aldrich, Milan, Italy) based on the dye JC-1 (5,5′,6,6′-tetrachloro-1,1′,3,3′-tetraethyl-benzimidazolocarbocyanine iodide) was used to measure changes in the mitochondrial inner membrane electrochemical potential gradient (Δψ) [17].

Aggregated JC-1 red fluorescence, representing intact mitochondria, and monomeric JC-1 green fluorescence of the disrupted mitochondria were detected at 525 nm (excitation)/590 nm (emission) and 490 nm (excitation)/530 nm (emission), respectively (Fluoroskan Ascent fluorimeter, ThermoLabsystems). A drop in the red/green fluorescence intensity ratio indicates mitochondrial depolarization.

#### 2.2.8. Caspase-3, Caspase-8 and Caspase-9

Caspase assays were performed by using specific caspase-3, 8 and 9 substrates, which release the fluorescence dye 7-amino-4-methylcoumarin (AMC, 380 nm excitation and 460 nm emission) upon their activation [16].

### 2.3. Experiments on Brain Slices

#### 2.3.1. Preparation of Cortical Slices

All animal care and experimental protocols used were approved by the Italian Department of Health (813/2015-PR) and agreed with the European Union Guidelines for the Care and the Use of Laboratory Animals (European Union Directive 2010/63/EU, http://ec.europa.eu/environment/chemicals/lab_animals/home_en.htm). The experimental details were already described [19,20]. Brains of male Wistar rats (250–300 g; Charles River Italia, Calco, Italy), were placed in 0.2 micron sterile filtered artificial cerebrospinal fluid (ACSF) (composition in mM): 120 NaCl; 2 KCl; 1 CaCl_2_; 1 MgSO_4_; 25 HEPES; 1 KH_2_PO_4_ and 10 glucose, pH 7.4, and cut into slices (400 µm, Stoelting Co., Wood Dale, IL, USA manual chopper) after dissecting the cortex [18]. Slices were left to recover from slicing trauma and then hydrogen peroxide-induced injury (10 mM H_2_O_2_ for 1 h), causing ~50% of tissue death, was performed [14].

#### 2.3.2. PRES Treatment

After recovering, slices were incubated with ACSF (controls) or ACSF + PRES (10–200 µg/mL for 1 h); PRES was then maintained and H_2_O_2_ (10 mM for 1 h) added. Slice viability along with tissue edema were determined at the end of the treatments. The effects of PRES per se were assessed in slices treated with the extract (10–200 µg/mL) for a corresponding period (2 h) [12].

#### 2.3.3. Brain Slices Viability Assays

The colorimetric MTT method [15,20], along with slices tissue edema (TE) [19] were used to check for tissue viability. Absorbance ((OD560 nm–OD630 nm) of untreated slices (controls) was taken as 100% of viability, while TE, given as g H_2_O (g dw)^−1^, was calculated according to the formula TE = (wet weight − dry weight)(dry weight)^−1^.

### 2.4. In Silico Prediction of Main PRES Component Pharmacokinetic Properties

The 2D Sketcher tool of Maestro (Schrödinger, LLC, New York, NY, USA, 2019) was used to build all the components of PREP. For every ligand, the protonation and tautomeric state, and their geometries, were optimized through LigPrep, part of the same suite. All the built structures were then submitted to the Qikprop program (Schrödinger Release 2019-2: QikProp, Schrödinger, LLC, New York, NY, USA, 2019) within Maestro to calculate the ADME properties and 2D and 3D QSAR descriptors.

### 2.5. Statistical Analysis

Data were collected as triplicate (SH-SY5Y cells) or quadruplicate (brain slices) from at least 4 independent experiments. The results are expressed as mean ± SEMs. Statistical significance was assessed by using one way or two way ANOVA, as appropriate (GraphPad Prism version 5.04, GraphPad Software Inc., San Diego, CA, USA). ANOVA assumptions were checked by the method of Bartlett (data sampled from populations with identical SDs) and Kolmogorov and Smirnov (data sampled from populations that follow Gaussian distributions). Statistical significance (*p*) was set at 0.05.

## 3. Results

### 3.1. Effects of PRES on SH-SY5Y Cell Viability

The effects of PRES per se (0–100 µg/mL for 24 h) on SH-SY5Y cells viability were evaluated to find suitable concentrations for assessing neuroprotective properties.

As reported in Figure 1, the extract was devoid of cytotoxic effects up to 50 µg/mL, while a significant reduction in cell viability (~28%) was detected at 100 µg/mL. This was confirmed also by phase-contrast light microscope examination of the cells, which highlighted grade 1 signs of cytotoxicity (i.e., ~20% of cells were round, tended to detach and did not show intracytoplasmic granules). Therefore, the neuroprotective effects of the extract were explored in the range 0–50 µg/mL.

### 3.2. PRES Reverted OS-Induced Decrease in SH-SY5Y Cells Viability

To reproduce OS, SH-SY5Y cells were treated with H_2_O_2_ (25 µM for 1 h followed by 24 h with medium). A decrease in cell viability of about 50% occurred upon the H_2_O_2_ challenge (Figure 2), while the 1 h pretreatment with PRES caused a concentration-dependent protection. Although not significantly, cells viability was partially recovered already at 1 µg/mL, while maximal neuroprotection was attained in the 5–50 µg/mL concentration range (25.8–34.1%, *p* < 0.01 vs. H_2_O_2_).

PRES neuroprotective effects were pointed out also by phase-contrast images. Compared to the control SH-SY5Y cells, those treated with H_2_O_2_ tended to round up and to detach from the well, showed shrinkage and membrane blebbing that are normally associated with apoptotic cell death (Figure 2 and Appendix A). PRES-treated cells instead appeared progressively healthy in both shape and number upon increasing PRES concentration.

### 3.3. PRES Reduced Apoptotic-Mediated SH-SY5Y Cell Death Caused by OS

Cell cycle analysis showed that after OS the percentage of sub G_0_/G_1_ hypodiploid, apoptotic cells was significantly higher (+12.0%, *p* < 0.001 vs. untreated cells), while that of G_0_/G_1_ phase decreased (−9.1%, *p* < 0.001 vs untreated cells) (Figure 3a). Interestingly, flow cytometry data also indicated that PRES (5–50 µg/mL) exerted neuroprotection, as the percentage of cell in sub G0/G1- and G_0_/G_1_-phase were brought back to basal values. Finally, the amounts of cells in S and G_2_/M phase were mostly unchanged in both OS and OS + PRES conditions. Nuclear apoptosis, assessed by using the florescent dye DAPI, confirmed previous results (Figure 3b).

H_2_O_2_ caused, in fact, a huge increase in the number of DAPI positive cells with fragmented and condensed nuclei, while in the presence of PRES a concentration-dependent reduction of these cells was observed, achieving the maximum effect at 5–50 µg/mL PRES.

The externalization of phosphatidyl serine (Annexin V/PI assay) was used to further assess PRES neuroprotection. As shown in Figure 4, the H_2_O_2_ challenge resulted in a marked increase in early and late apoptotic cells, as well as in those necrotic, while the population of viable cells dropped.

In contrast, PRES-pretreated SH-SY5Y cells showed significant resistance to OS-induced cytotoxicity, especially when using the natural extract in the 5–50 µg/mL concentration range. In this case, viable, early apoptotic, and necrotic cells gradually returned to control values upon increasing PRES concentration, while the late apoptotic population seemed not to be affected.

### 3.4. PRES Hampered the Formation of ROS Along with Loss in Mitochondria Membrane Potential Caused by OS

PRES lowered in a concentration- and time-dependent pattern intracellular ROS formation caused by H_2_O_2_ (Figure 5a), as highlighted by DCF-DA assay. Significant protection already occurred at 1 µg/mL after 24 h, while maximum effect was achieved at 10–50 µg/mL at both tested times. Finally, PRES per se had negligible effects on ROS production.

The mitochondrial membrane potential (Ψm) is closely linked to functional activity and loss of Ψm (depolarization) is an initial sign of apoptosis, being a result of mitochondrial uncoupling. The change in JC-1 fluorescence from red to green was used to assess Ψm changes. ΔΨm of hydrogen-peroxide-treated cells was reduced by 36% after 24 h (Figure 5b). PRES restored Ψm values and the maximal effect was achieved at 10 µg/mL. These results suggest that PRES preserves mitochondrial function, thus probably hampering the initiation of mitochondrial-dependent apoptosis.

### 3.5. The Increase in Caspase Activity Caused by OS Was Reduced by PRES

Caspase-9 and -8 are the key enzymes of the apoptotic mitochondria- and the extrinsic receptor-mediated pathways, respectively. These, in turn, activate caspase-3 [21]. To investigate whether PRES could revert the activation of these apoptotic pathways caused by OS, specific fluorescent caspase-3, -8 and -9 substrates were used. As 10 and 50 µg/mL were equally active in the previous assays, PRES was tested in the range 1–10 µg/mL. The effects of PRES per se (10 µg/mL) were also assessed.

Results showed that the activity of both cleaved caspase-3 and -9, but not that of caspase-8, were significantly increased upon OS (Figure 6). The activation of the mitochondrial pathway was significantly reverted by PRES at 1 or 10 µg/mL as both caspase-3 and caspase-9 activity regained basal values. Finally, PRES per se did not affect caspase-3, -8 and -9 activity.

### 3.6. PRES Neuroprotection Occurred Also in Rat Brain Slices Subjected to OS

PRES neuroprotection was also assessed in rat brain slices, in which the main structural and synaptic organization of the original tissue is conserved [22]. As a first step, the effects of PRES per se (0–200 µg/mL) on brain slices viability (MTT assay) and tissue edema were evaluated. The extract was devoid of cytotoxic properties in the range of the tested concentrations (Figure 7a,b): however, as a slight, not significant reduction in slices viability was observed upon 200 µg/mL, PRES neuroprotective activity was tested in the range of 0–100 µg/mL. For these experiments, PRES pretreatment (1–100 µg/mL, 1 h) was performed, followed by an OS challenge (H_2_O_2_ 1 mM, 1 h): afterward, slice viability and tissue edema were assessed. OS caused a significant reduction in slice viability (54.1 ± 2.6%, *p* < 0.001), which was accompanied by an increase in tissue edema (54.12 ± 15.6%, *p* < 0.001) (Figure 7c,d). Interestingly PRES reverted the effects of OS at 50 and 100 µg/mL, while 10 µg/mL was ineffective.

### 3.7. In Silico Prediction of PRES Component’s Drug-Like Properties

As the neuroprotective potential of PRES is strictly related to its penetration into the brain and a relatively good pharmacokinetic profile, an in-silico prediction of PRES component’s drug-like properties was carried out (Table 1). Moreover, oleuropein has been assessed either as glycoside or as aglycone, being the latter the main form assimilated in the shape of oleuropein, and endowed of biological and pharmacological properties, including a remarkable antioxidant activity.

In particular, quantitative structure-activity relationship studies, considering CNS-active drugs, have suggested a set of physicochemical parameters that CNS compounds should possess. They are molecular weight <450 Dalton, a number of hydrogen bond acceptors <7, a polar surface area <90 Å, a ClogP <5, and a number of torsions ≤10. Considering these features, oleuropein has a good probability of entering the CNS (Table 1), considering that the majority of the predicted values fall within the above-described ranges. Moreover, the QPPMDCK value, which gives an estimation of the diffusion into MDCK cells (models of the BBB), shows that oleuropein would have adequate brain diffusion features. This is further supported by the predicted brain/blood partition coefficient for orally administered drugs (QPlogBB). Interestingly the same component is also predicted to overcome the gut/blood barrier (see QPPCaco in Table 1) thereby supporting its oral bioavailability. Apart from oleuropein, the other components are not predicted to easily cross the BBB except for elenolic acid glucoside and hibiscus acid, for which the QPlogBB (Table 1) indicates the potential to be brain-penetrant. Overall, from these calculations, oleuropein, elenolic acid glucoside, and hibiscus acid are the three PRES components that would show the highest possibility to exert their biological functions at CNS.

## 4. Discussion

The present work has assessed neuroprotective properties of PRES toward OS-mediated injury in both human neuroblastoma SH-SY5Y cells and in rat brain slices. OS-induced injury was selected as it is strictly involved in the mechanisms linked to neurodegeneration, either as a cause or as a crucial factor of the downstream cascade which leads to neuronal death [23]. The present findings demonstrated that OS caused by H_2_O_2_ challenge induced ROS production accompanied by apoptotic death of SH-SY5Y cells, shown as an increase in hypodiploid cells (cell cycle analysis), early and late apoptotic cells (AV/PI staining), as well as in chromatin condensation and DNA fragmentation (DAPI staining). The fluorescent probe JC-1 was also used to investigate the effects of OS on mitochondria membrane potential.

A membrane depolarization occurred, strongly suggesting the triggering of the apoptotic intrinsic, mitochondrial-mediated pathway, that involves the activation of the downstream caspase-9, which leads to active caspase-3 and -7 [21]. The results showed an increase in caspase-9 and caspase-3, but not caspase-8, activity following H_2_O_2_ treatment, thus confirming this hypothesis. The present findings agree with other reports on the same cell line [24,25], but not with [26], which reported caspase-8-mediated extrinsic pathway activation, and [27], which described both intrinsic and extrinsic involvement. This discrepancy can be explained by considering the different concentrations and times of exposure of H_2_O_2_ used to induce OS, the types of assays and different parameters used in each study to assess the injury, as already examined by [27].

Results here reported demonstrated that PRES may have potential in preventing OS-mediated neuronal injury since it can revert loss in cell viability (MTT assay). When neurons are exposed to oxygen and nitrogen species, a huge amount of energy is required to protect tissues from the stress. This causes mitochondrial malfunction, with the release of cytochrome C and other mitochondrial proteins, which in turn drive apoptosis [28]. PRES reverted the formation of ROS and apoptotic cells, the loss in MPP as well as caspase-3 and caspase-9 activation. Interestingly, PRES cytoprotective and antioxidant effects occurred at the same concentrations (5–50 µg/mL) in both SH-SY5Y and HUVEC cells exposed to the same challenge [8].

To better understand the mechanism underlying PRES neuroprotection, it is crucial to refer to its main components. PRES composition, characterized by HPLC coupled to a UV-Vis and QqQ-Ms detector, has already been reported [8,9] and the main components, fully detailed in Materials and Methods, consist in elenolic acid glucoside (23.65 ± 0.79 mg/g), oleuropein (215.1 ± 1.64 mg/g), oleuropein isomer (51.09 ± 0.35 mg/g) for *Olea europaea* L. leaves, and hibiscus acid (139.2 ± 0.47 mg/g) for *Hibiscus sabdariffa* L. calyces. These compounds possess antioxidant properties [29,30,31] and this could be at the basis of the ability of PRES to revert ROS formation and thus to elicit neuroprotection. Numerous in vitro and in vivo reports have proven oleuropein-derived numerous positive effects in consideration of its antioxidant, antimicrobial, antiviral, hepatoprotective, antitumoral, antiaging, cardioprotective, and anti-inflammatory properties [29,32]. In terms of neuroprotection, the contributions of oleuropein are its dose-dependent, strong, antioxidant property linked to metal ions chelating activity, the ability to scavenge nitric oxide, to decrease ROS and RNS, and to reduce lipid peroxidation [33]. Also its antiapoptotic properties as well as the ability to activate the intracellular ubiquitin/proteasome pathway, which drive the degradation of damaged or misfolded proteins, have been proven to contribute [33,34]. Inhibition of inflammatory enzymes, such as lipoxygenases, and reduction of inflammatory mediators (TNF-α, NF-kB, IL-1β and IL-6) have been also described [32]. Being oleuropein and its isomer the most representative from a quantitative point of view, it is reasonable to hypothesize that PRES-mediated neuroprotection is mainly driven by oleuropein itself. Interestingly, however, the most recent experimental studies in SH-SY5Y cells subjected to different injuries have shown that oleuropein exert neuroprotection at concentrations 20–50 µM (see, for example, [35,36]). The present results showed that this compound is active at 2–4 µM, thus suggesting a contribution of other components of PRES. Among these, elenolic acid glucoside and hibiscus acid, the most representative from a quantitative point of view after oleuropein, might play a role. While the neuroprotective activity of elenolic acid glucoside is elusive, that of hibiscus acid is instead known. This compound was proven to ameliorate cognitive dysfunction in vivo via antioxidant effect, to boost the cholinergic system, to inhibit tau hyperphosphorylation pathways by regulating the expression of *p*-JNK, *p*-tau, and c-PARP [37]. Moreover, in a mice model of multiple sclerosis, hibiscus acid is neuroprotective by ameliorating inflammation and OS [38]. Taken together we can conclude that PRES neuroprotective activity toward OS-induced SH-SY5Y apoptotic death, is due to the antioxidant, as well as antiapoptotic, properties of its components, which might act synergistically. This is further supported by the observation that in HUVEC cells cytoprotective and antioxidant properties of the single extracts from *Olea europaea* L. leaves and *Hibiscus sabdariffa* L. calyces were less active in reverting OS-mediated injury than the 13:2 (*w*/*w*) mixture (i.e., PRES used in the present study) [8].

SH-SY5Y cells results were further supported by those obtained in rat brain slices, an in vitro experimental model that maintains functional synapses along with three-dimensional tissue architecture, and widely used to assess neuroprotection [39]. PRES still exerted protection, although at higher concentrations than in SH-SY5Y cells (50–100 µg/mL). This can be explained by considering that in brain slices, target(s) site(s) is(are) reached after PRES permeation across the membranes and this could require higher concentration in comparison to cultured cells, where the targets are certainly more accessible.

In the case of neuroprotective compounds, the ability to access the brain is a crucial point. For these reasons, we have performed an in-silico prediction of PRES component’s drug-like properties. These studies confirmed that among all the PRES components, oleuropein has the highest chance to be orally bioavailable and cross the blood-brain barrier. Additionally, to some extent, elenolic acid glucoside and hibiscus acid would have a comparable chance of distributing in the brain. Nevertheless, it should be also outlined that the ADME properties of this nutraceutical product, and consequently its biological effect in vivo, might be ascribed to the concurrent presence of all the PRES components. Finally, both oleuropein and hibiscus acid have shown interesting and promising neuroprotective effects also in in vivo models of neurodegenerative diseases [10,40,41,42,43]. Moreover, oleuropein or its active metabolite, hydroxytyrosol, is found in a significant amount in the brain following oleuropein peripheral administration [44,45]. These observations, along with in silico data of the present study, support the possibility of the PRES main component to cross the blood-brain barrier, supporting its use as a neuroprotective agent.

## 5. Conclusions

Effective treatments for neurodegenerative disorders represent a significant challenge for medicine. Preventive strategies can decrease not only the risk of developing the disease, but also the progression of their symptoms [46]. A possible approach is to use a multi-target therapy, based on the use of compounds able to act simultaneously on more than one target, with the final goal of achieving a synergistic action and, thus, a better therapeutic efficacy. Natural derivatives constitute an important source of multi-target compounds, and these might drive toward a novel, interesting direction the discovery of novel compounds for delaying the onset or the progression of neurodegenerative disorders [47]. Their importance is further supported by the synergistic effect often caused by the combination of the different components [48,49]. Intake of phytochemicals on a regular basis might in fact boost the antioxidant system, thus increasing neuronal cell survival and improving physical and mental activity [47,50]. The present findings highlight interesting neuroprotective properties of PRES, probably due to the multi-target, synergic effects of its main components. The possible good penetration in the brain constitutes an added value to its potential use as a nutraceutical, which could help in preventing neurodegenerative diseases.

## Figures and Tables

**Figure 1 antioxidants-09-00806-f001:**
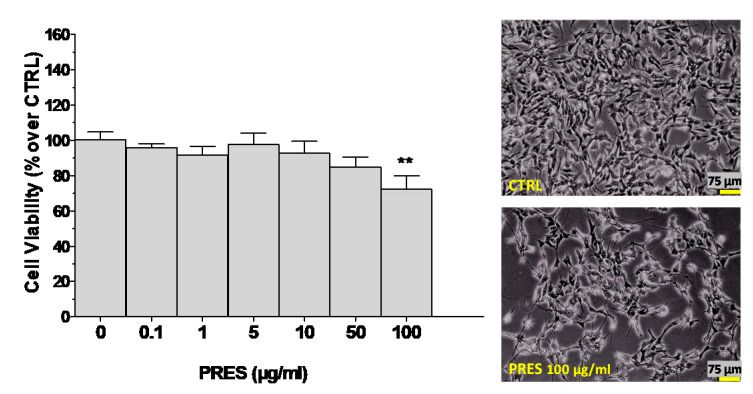
Effects of *Pres phytum* (PRES) on SH-SY5Y cell viability. The cells were incubated with increasing concentrations of PRES (0–100 µg/mL, 24 h) and an MTT test was performed. Data are reported as mean ± SEMs. ** *p* < 0.01 vs. untreated cells (one-way ANOVA and Bonferroni post-test). Upper panels: Morphological analysis at contrast phase microscopy between untreated (CTRL) and PRES-treated (100 µg/mL for 24 h) SH-SY5Y cells (scale bar 75 µm) was performed according to USP 28 (United States Pharmacopeia edition 2005) grade scale for assessment of the cytotoxic potential of tested materials. Each photograph was representative of 3 independent observations.

**Figure 2 antioxidants-09-00806-f002:**
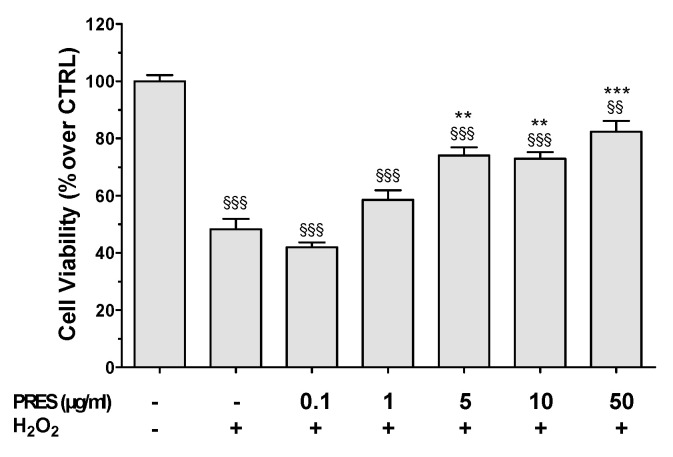
Effects of PRES on oxidative stress (OS)-induced cytotoxicity in SH-SY5Y cells. Cells were treated with PRES (1–50 µg/mL) for 1 h before OS and for the following 24 h. OS was reproduced by using H_2_O_2_ (25 µM for 1 h + 24 h with medium). Cell viability was assessed by MTT assay and data are reported as means ± SEMs. §§ *p* < 0.01, §§§ *p* < 0.001 vs. untreated cells; ** *p* < 0.01, *** *p* < 0.001 vs. H_2_O_2_ (one-way ANOVA and Bonferroni post-test).

**Figure 3 antioxidants-09-00806-f003:**
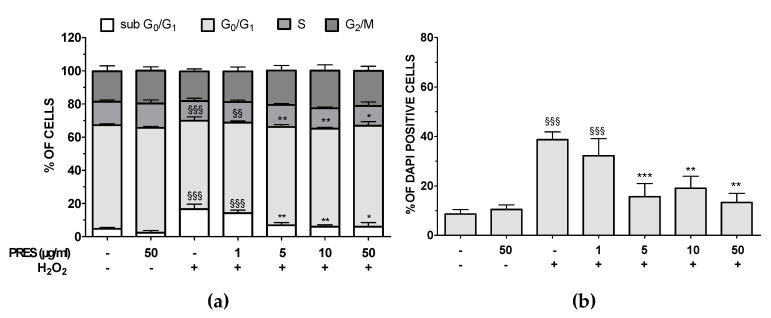
Effects of PRES on OS-induced changes in SH-SY5Y cell cycle and DNA condensation. Cells were treated with PRES (1–50 µg/mL) for 1 h before OS and for the following 24 h. OS was reproduced by using H_2_O_2_ (25 µM for 1 h + 24 h with medium). (**a**): cell cycle analysis: percent of cells in the subG_0_/G_1_ (apoptotic), G_0_/G_1_, S or G_2_/M phase determined by flow cytometry after propidium iodide staining. (**b**): apoptotic cells detected by DAPI staining: quantitative analysis. Data are reported as means ± SEMs. §§ *p* < 0.01, §§§ *p* < 0.001 vs. untreated cells; * *p* < 0.05, ** *p* < 0.01, *** *p* < 0.001 vs. H_2_O_2_ (one-way ANOVA and Bonferroni post-test).

**Figure 4 antioxidants-09-00806-f004:**
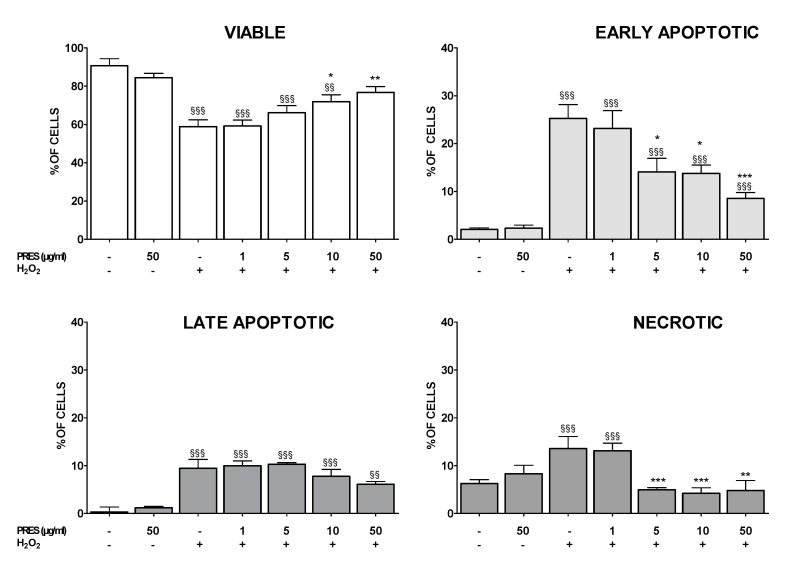
Effects of PRES on OS-induced apoptosis in SH-SY5Y cells. Cells were treated with PRES (1–50 µg/mL) for 1 h OS and for the following 24 h. OS was reproduced by using H_2_O_2_ (25 µM for 1 h + 24 h with medium). At the end of the treatment, flow cytometric analysis of cells stained with Annexin V and propidium iodide (PI) was performed. Viable: cells negative for both PI and AV. Early apoptotic: cells positive for annexin V and negative for PI. Late apoptotic: cells positive for both PI and AV. Necrotic: positive for PI and negative for AV. Data are reported as means ± SEMs. §§ *p* < 0.01, §§§ *p* < 0.001 vs. untreated cells; * *p* < 0.05, ** *p* < 0.01, *** *p* < 0.001 vs. H_2_O_2_ (one-way ANOVA followed by Bonferroni post-test).

**Figure 5 antioxidants-09-00806-f005:**
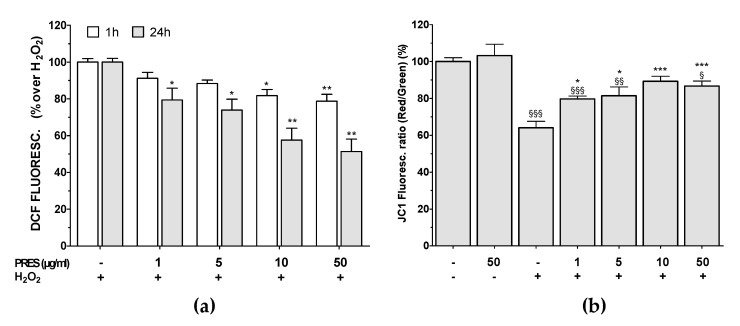
Effects of PRES on OS-induced ROS and ΔΨm changes in SH-SY5Y cells. (**a**): ROS were evaluated by the oxidation of DCF-DA to DCF (**b**): JC-1 dye was used as a probe for changes of ΔΨm. The ratio between red/orange- (aggregates, 527 nm) and green- (monomers, 590 nm) emissions was calculated and reported as percent vs. untreated cells. Data are reported as means ± SEMs and analyzed by two-way (DCF) or one-way (JC1) ANOVA. § *p* < 0.05, §§ *p* < 0.01, §§§ *p* < 0.001 vs. untreated cells; * *p* < 0.05, ** *p* < 0.01, *** *p* < 0.001 vs. H_2_O_2_.

**Figure 6 antioxidants-09-00806-f006:**
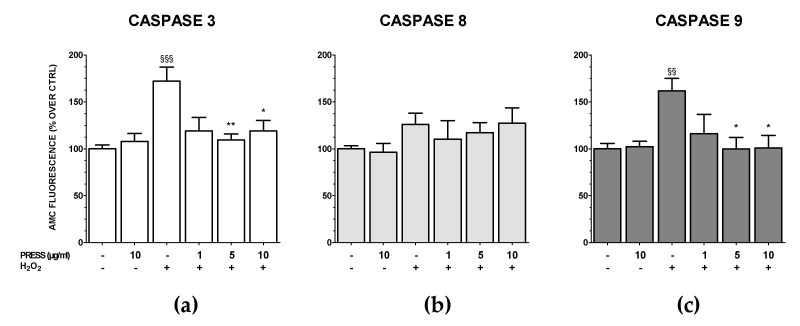
PRES-mediated effects on OS-induced changes in caspase-3, -8 and -9 activity in SH-SY5Y cells. Specific substrates for caspase-3 (a), caspase-8 (b) and caspase-9 (c), which release the fluorescence probe AMC (380 nm exc and 460 nm em) when activated, were used. Data are reported as mean ± SEMs. §§ *p* <0.01, §§§ *p* < 0.001 vs. untreated cells; * *p* < 0.05, ** *p* < 0.01 vs. H_2_O_2_ (ANOVA followed by Bonferroni post-test).

**Figure 7 antioxidants-09-00806-f007:**
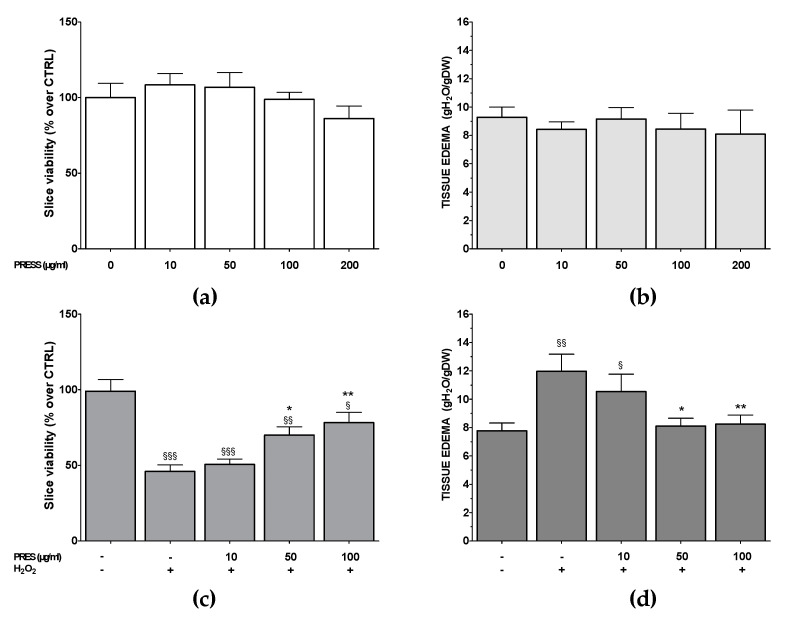
Effects of PRES on OS-induced reduction in rat brain slices viability and water content (tissue edema). (**a**,**b**): effects of PRES per se (10–200 µg/mL, 2 h) on tissue viability (MTT assay) and edema, respectively. (**c**,**d**): Slices were incubated with artificial cerebrospinal fluid (ACSF) (controls) or ACSF + PRES (10–100 µg/mL for 1 h). Afterward, PRES was maintained H_2_O_2_ (10 mM for 1 h) was added. Data are reported as means ± SEMs. §§§ *p* < 0.001, §§ *p* < 0.01, § *p* < 0.05 vs. untreated slices; * *p* < 0.05, ** *p* < 0.01 vs. H_2_O_2_ (ANOVA followed by Bonferroni post-test).

**Table 1 antioxidants-09-00806-t001:** Predicted physico-chemical parameters for all the PRES components.

Calculated Parameters	Compounds										Range of Recommended Values *^a^*
	Rutin	Elenolic Acid Glucoside	Hydroxyoleuropein	Hibiscus Acid	Verbascoside	Ligstroside	Oleuropein	Oleurpoein Aglycone	Luteolin-7-O-Rutinoside	Luteolin-7-O-Glucoside	
**#rotor *^b^***	15	10	17	3	20	14	15	9	16	10	0–15
**Lipinski Rule of 5 violations *^c^***	3	1	3	0	3	2	3	0	3	2	N.A.
**mol_MW *^d^***	610.5	404.4	556.5	190.1	624.6	524.5	540.5	378.4	610.5	448.4	130.0–725.0
**dipole *^e^***	10.3	5.0	7.9	4.1	4.7	5.5	9.6	5.2	10.8	6.7	1.0–12.5
**SASA *^f^***	895.4	622.9	824.8	333.6	924.4	801.8	808.6	612.8	820.5	702.0	300.0–1000.0
**donorHB *^g^***	9.0	5.0	7.0	2.0	9.0	5.0	6.0	3.0	9.0	6.0	0.0–6.0
**accptHB *^h^***	20.6	15.9	19.1	6.8	20.3	16.7	17.4	8.9	21.5	13.0	2.0–20.0
**QPlogPo/w *^i^***	−2.4	−1.3	−1.5	−0.8	−1.5	0.0	−0.5	1.0	−2.7	−1.0	−2.0–6.5
**QPlogS *^j^***	−3.3	−1.8	−2.4	−0.6	−2.8	−3.0	−2.8	−3.0	−2.1	−3.2	−6.5–0.5
**QPPCaco *^k^***	0.3	6.7	3.6	1.4	1.0	29.6	12.5	68.0	0.9	3.6	<25 poor, >500 great
**QPlogBB *^l^***	−5.7	−2.5	−4.3	−1.7	−5.5	−3.1	−3.5	−2.1	−4.8	−3.8	−3.0–1.2
**QPPMDCK *^m^***	0.1	2.8	1.1	0.7	0.3	11.0	4.3	27.1	0.3	1.1	<25 poor, >500 great
**Jorgensen Rule of 3 violations *^n^***	2	2	2	1	2	1	2	0	2	2	N.A.
**#metab *^o^***	10	7	10	3	10	9	10	6	10	7	1–8

*^#^* numbers of *^a^* For 95% of known drugs. *^b^* Number of non-trivial (not CX3), non-hindered (not alkene, amide, small ring) rotatable bonds. *^c^* Predicted numbers of violations of Lipinski’s rule of five. *^d^* Molecular weight of the molecule. *^e^* Computed dipole moment of the molecule. *^f^* Total solvent accessible surface area (SASA) in square angstroms using a probe with a 1.4 Å radius. *^g^* Estimated number of hydrogen bonds that would be donated by the solute to water molecules in an aqueous solution. *^h^* Estimated number of hydrogen bonds that would be accepted by the solute from water molecules in an aqueous solution. *^i^* Predicted octanol/water partition coefficient. *^j^* Predicted aqueous solubility, log S. S in mol dm^−3^ is the concentration of the solute in a saturated solution that is in equilibrium with the crystalline solid. *^k^* Predicted apparent Caco-2 cell permeability in nm/sec. Caco-2 cells are a model for the gut-blood barrier. *^l^* Predicted brain/blood partition coefficient for orally delivered drugs. *^m^* Predicted apparent MDCK cell permeability in nm/s. MDCK cells are considered to be a good mimic for the blood-brain barrier. *^n^* Predicted numbers of violations of Jorgensen rule of three. *^o^* Number of likely metabolic reactions. Values falling in the recommended ranges are highlighted in green.

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
