# Peer review of "Olive Leaves and Hibiscus Flowers Extracts-Based Preparation Protect Brain from Oxidative Stress-Induced Injury"

_antioxidants, 2020, doi:10.3390/antiox9090806_

Round 1

Reviewer 1 Report

In this study, titled “Hibiscus flowers and olive leaves extracts-based formulation protect brain from oxidative stress-induced injury”, the authors investigated the ability of PRES to induce neuroprotection of SH-SY5Y cell cultures and in acute brain slices. The data presented here suggests that PRES is able to limit neuronal cell damage induced by oxidative stress, by decreasing apoptosis, necrosis, and with preventing mitochondria membrane depolarization. Overall, this study suggest that PRES induces neuroprotection in vitro models of oxidative stress, however, this review has a few comments regarding this study:

  • In line 129, the authors state that they followed the manufacturer’s protocol, with slight modifications. What were these modifications?
  • Line 135, the authors appear to have forgot to list a reference “(REF)”.
  • Why was this composition of ACSF used (line 185-186)? Specifically, why was the Na ion concentration lower then what is typically in ACSF, as ~26mM NaHCO3 is commonly found in ACSF.
  • Line 189, this citation does not have any mention of slices or ACSF, why is this citation here?
  • Figure 5a, why did the authors not include a negative/negative control (PRES/H2O2) and PRES 50/Neg in this figure, like that found in 5b?
  • Line 365, I believe this passage refers to Figure 7 and not Figure 6. Please confirm.
  • As the authors state, the neuroprotective properties of PRES is strictly related to its penetration into the brain. Based upon their own data and as they mention, the only compound that has a capacity to penetrate the brain is oleuropein. (as they state, all other components are not predicted to easily cross the BBB.) Why wasn’t the focus of the paper on oleuropein, since the data presented here was not focused on oleuropein and PRES-induced may be from other components?
    • Or compared PRES with oleuropein, elenolic acid glucoside, and hibiscus acid together to determine the potential for synergistic neuroprotection.

Author Response

In this study, titled “Hibiscus flowers and olive leaves extracts-based formulation protect brain from oxidative stress-induced injury”, the authors investigated the ability of PRES to induce neuroprotection of SH-SY5Y cell cultures and in acute brain slices. The data presented here suggests that PRES is able to limit neuronal cell damage induced by oxidative stress, by decreasing apoptosis, necrosis, and with preventing mitochondria membrane depolarization. Overall, this study suggest that PRES induces neuroprotection in vitro models of oxidative stress, however, this review has a few comments regarding this study:

  • In line 129, the authors state that they followed the manufacturer’s protocol, with slight modifications. What were these modifications?

DAPI (4’,6-diamidino-2-phenylindole) kit manufacturer's protocol suggest to use 0.3 µM DAPI for 5 min for the staining, followed by 3 whashes in PBS (https://www.thermofisher.com/it/en/home/references/protocols/cell-and-tissue-analysis/protocols/dapi-imaging-protocol.html). Our protocol used 0.5 µM DAPI for 10 min, followed by 2 washed in PBS and 1 with water. These details are now added in the text (see lines 144-145 ms tracked version)

  • Line 135, the authors appear to have forgot to list a reference “(REF)”.

We apologize for the typo: the sentence has been rearranged properly (see line 151, ms tracked version).

  • Why was this composition of ACSF used (line 185-186)? Specifically, why was the Na ion concentration lower then what is typically in ACSF, as ~26mM NaHCO3 is commonly found in ACSF.

The ACS used for performing experiments with acute rat brain slices often contain slight changes in its composition. This is essentially made to improve slice health in terms of energy metabolism, morphology, electrophysiological responsiveness etc. In our experience, slices tend to take up water and thus to become edematous in normal ACSF, in agreement with data of the literature (see PMID: 11323105), while when using the reported “HEPES-ACSF” (in mM: 120 NaCl; 2 KCl; 1 CaCl2; 1 MgSO4; 25 HEPES; 1 KH2PO4 and 10 glucose), tissue viability is much improved. This ACSF composition has already widely used by other Authors (see for ex. PMID 24412540, 28506823, 29506554) and by other studies of our group (see for ex PMID 32470081, 27739104, 27923201, 26942555 and others).

  • Line 189, this citation does not have any mention of slices or ACSF, why is this citation here?

We apologize for the mistake; the reference has been now properly positioned (see line 148 ms tracked version)

  • Figure 5a, why did the authors not include a negative/negative control (PRES/H2O2) and PRES 50/Neg in this figure, like that found in 5b?

In Figure 5a, data were expressed as the percent of inhibition of intracellular DCF produced by H2O2-exposure (PMID 27739104, 26240013, 23533692). The effects of PRES 50 µg/ml “per se” on ROS production, which were negligible at both selected time points, were not reported as the scale of Y-axis was above its value. We agree with the Referee that this information might be useful, and this has been now stated in the text (see lines 350-1 ms tracked version).  

  • Line 365, I believe this passage refers to Figure 7 and not Figure 6. Please confirm.

Yes, the Referee is right. The correct figure to be quoted is fig 7 and not 6. This has been now properly amended in the text (see line 357 ms tracked version).

  • As the authors state, the neuroprotective properties of PRES is strictly related to its penetration into the brain. Based upon their own data and as they mention, the only compound that has a capacity to penetrate the brain is oleuropein. (as they state, all other components are not predicted to easily cross the BBB.) Why wasn’t the focus of the paper on oleuropein, since the data presented here was not focused on oleuropein and PRES-induced may be from other components? Or compared PRES with oleuropein, elenolic acid glucoside, and hibiscus acid together to determine the potential for synergistic neuroprotection. 

In silico prediction of PRES component’s drug-like properties suggested that beside oleuropein, both elenolic acid glucoside and hibiscus acid, have the potential to be brain-penetrant in consideration of the predicted brain/blood partition coefficient for orally delivered drugs (QPlogBB in Table 1)(see lines 436-7 ms tracked version). This has been confirmed by in vivo neuroprotective activity of hibiscus acid (see lines 529-32 ms tracked version), while the in vivo properties of elenolic acid glucoside are still more elusive. The present study was aimed at assessing neuroprotective properties of PRES by using in vitro approach (cells and rat brain slices subjected to oxidative stress). Results suggested that oleuropein mainly drive neuroprotection and being its concentrations 10-20 fold lower than those reported to be active in the same cells in the literature, the involvement of other component of the extract might be advanced. Synergistic activity of these components was already reported in HUVEC cells, as cytoprotective and antioxidant properties of the single extracts from Olea europaea L. leaves and Hibiscus sabdariffa L. calyces were less active in reverting OS-mediated injury than the 13:2 (w/w) mixture (i.e. PRES used in the present study) (PMID: 26180582). Thus this study can be considered a “preliminary” assessment of the potential of PRES, as we are planning to verify neuroprotective properties of PRES in in vivo models, in which we hope to also confirm the synergistic effects of the main PRES components. Finally, it worth to remind that, although from the in silico analysis, oleuropein, elenolic acid glucoside, and hibiscus acid are the three PRES components that would show the highest possibility to exert their biological functions at CNS, ADME properties of this nutraceutical product, and consequently its biological effect in vivo, might be dependent to the concurrent presence of all the PRES components. This point has been further stressed in the text (see lines 438-440 and 549-551, ms tracked version).

Reviewer 2 Report

In this manuscript, Chiaino and collegues report the neuroprotective role of PRES, a nutraceutical product composed of leaves- and flowers-extracts of Olea europaea 24 L. and Hibiscus sabdariffa L., in human neuroblastoma SH-SY5Y cells and in rat brain slices subjected to oxidative stress. The Authors clearly demonstrate the antioxidant properties of PRES and show its potential use in preventing neurodegenerative disorders. Also, in silico data demonstrate that PRES main components exhibit a possible good penetration of the blood-brain barrier which makes it suitable as neuroprotective agent.

Overall, the authors make a convincing case that PRES is neuroprotective against oxidative stress. However, before this study is suitable for publication, the authors must address some minor issues listed below.

A general comment regards the fact that the cellular model used (SH-SY5Y) allows to study the PRES neuroprotective property against oxidative stress in different neuronal types. Indeed, the undifferentiated SH-SY5Y cells display a dopaminergic phenotype, whereas the retinoic acid-differentiated cells show a cholinergic phenotype. These two cell types show a different sensitivity towards potentially neuroprotective substances. Please see Branca JJV et al., Selenium and zinc: Two key players against cadmium-induced neuronal toxicity. Toxicol In Vitro. 2018 Apr;48:159-169.doi: 10.1016/j.tiv.2018.01.007 and comment.

Minor comments:

1) Page 2 line 46 please substitute .” ROS, in fact, …” with “Indeed, ROS…”.

2) Page 2 line 57 please add “to” between “due” and “the”

3) In 3.2 section of results (page 6 line 249) the Authors describe the H2O2-dependent morphological changes. However the figures magnification is too low to appreciate these changes and the PRES-dependent reversion. It would be better to show inserts aat higher magnification where these changes are highlighted.

4) In the discussion section (page 12 line 450) the Authors point out that their results are superimposable with that obtained in HUVEC cells. They should add a comment as to why they think these results are interesting. Also, please rephrase the sentence between line 462 and 467 because it is too long and complicated.

5) Page 13 line 482 please add “to” between “due” and “the”.

Author Response

A general comment regards the fact that the cellular model used (SH-SY5Y) allows to study the PRES neuroprotective property against oxidative stress in different neuronal types. Indeed, the undifferentiated SH-SY5Y cells display a dopaminergic phenotype, whereas the retinoic acid-differentiated cells show a cholinergic phenotype. These two cell types show a different sensitivity towards potentially neuroprotective substances. Please see Branca JJV et al., Selenium and zinc: Two key players against cadmium-induced neuronal toxicity. Toxicol In Vitro. 2018 Apr;48:159-169.doi: 10.1016/j.tiv.2018.01.007 and comment.

The immortalized and proliferative cell line SH-SY5Y is one of the most commonly used in vitro experimental models not only for Parkinson research, but also for Alzheimer’s disease, neurotoxicity, ischemia or Amyotrophic Lateral Sclerosis (see for ex. what reported by PMID 28118852). For a screening projects, such as the present study, the availability of large amounts of phenotypically homogenous cells with identical genetic background is crucial, especially to easily set up the experimental design, for data interpretation and comparison with those of the Literature. A further potential advantage of SH-SY5Y cells is the possibility for neuronal differentiation and several protocols have been developed (PMID 23975817), which should further drive the neuronal phenotype towards a dopaminergic one. Indeed, this is still controversial, and many Authors also reported a cholinergic phenotype, as outlined by the Referee. Moreover, the various protocols used for differentiation make SH-SY5Y cells differently responsive to the common toxic injury and to neuroprotectants, being these effects closely related to the neuronal phenotype, as outlined also by the paper quoted by the Referee. The change of susceptibility appears to be dependent upon the differentiating agents used and not upon differentiation “per se” (PMID 20497720). In consideration of the complexity and time-consuming procedures to differentiate SHSY-5Y cells, to characterize them and considering the possible different response of neuroprotectants, we prefer to validate and further understand the undifferentiated cellular results in a more physiological context such as rat brain slices. As delineated in the manuscript, these preserve much of the complex cellular integrity, including cellular barriers and intact circuitry, and as a result conserve functionality, resulting in an in vitro environment much comparable to the in vivo brain.  Giving the present results, which can be considered a “preliminary” assessment of the potential of PRES, we are planning to verify neuroprotective properties of PRES in in vivo models, in which we hope to also confirm the synergistic effects of the main PRES components.

Minor comments:

1) Page 2 line 46 please substitute .” ROS, in fact, …” with “Indeed, ROS…”.

The text has been properly amended (see line 46 ms tracked version).

2) Page 2 line 57 please add “to” between “due” and “the”

The text has been properly amended (see line 60 ms tracked version).

3) In 3.2 section of results (page 6 line 249) the Authors describe the H2O2-dependent morphological changes. However the figures magnification is too low to appreciate these changes and the PRES-dependent reversion. It would be better to show inserts aat higher magnification where these changes are highlighted.

We agree with Referee that figure magnification is not sufficient to appreciate PRES-induced changes. For this reason, we have moved photos of figure 2 also to supplementary info, in which a higher magnification is possible.

4) In the discussion section (page 12 line 450) the Authors point out that their results are superimposable with that obtained in HUVEC cells. They should add a comment as to why they think these results are interesting. Also, please rephrase the sentence between line 462 and 467 because it is too long and complicated.

We agree with Referee that the sentence does not sufficiently highlight the fact that PRES-mediated protection occur in the same concentration range also in a different cell line. Thus, it has been changed into: “Interestingly, PRES cytoprotective and antioxidant effects occurred at the same concentrations (5-50 µg/ml) in both SH-SY5Y and HUVEC cells exposed to the same challenge [7]” (see lines 501-3 ms tracked version).

The sentence of line 462-467 has been rearranged as follows:

In terms of neuroprotection, contribute to oleuropein effects its dose-dependent, strong, antioxidant property linked to metal ions chelating activity, the ability to scavenge nitric oxide, to decrease ROS and RNS, and to reduce lipid peroxidation [32]. Also its anti-apoptotic properties as well as the ability to activate the intracellular ubiquitin/proteasome pathway, responsible for the degradation of misfolded or damaged proteins have been proven to contribute [32, 33](see lines 513-19 ms tracked version).

5) Page 13 line 482 please add “to” between “due” and “the”.

The text has been properly amended (see line 533 ms tracked version).

Reviewer 3 Report

In this manuscript entitled "Olive leaves and hibiscus flowers extracts-based preparation protect brain from oxidative stress-induced injury", Chiaino and colleagues aimed at evaluating the neuroprotective effect of PRES in an in vitro model of human 'neuronal-like' cells and in rat brain slices.

Here listed some concerns and suggestions to improve the manuscript:

  1. Statistical considerations: please insert a section entitled 'Statistical analysis' or 'Statistical consideration' instead of 'Analysis of Data'. Did the authors check data for normality? Which test was used?
  2. For comparisons between group authors used ANOVA followed by Bonferroni post(-hoc) test. This analysis of variance is appropriate for comparisons between n>2 groups and 1 independent variable. In almost all the experiments you have 2 independent variables (H2O2 and PRES treatment), which means that multiway ANOVAs is required.
  3. Another issue is related to statistical comparisons between treated groups and control. Eg – Figure 2: Is the cell viability of PRES 0.1 ug/ml group significantly reduced as compared to control (no PRES no H2O2)?
  4. Figure 3, figure 4, figure 5, figure 7 are all example of neglected comparisons. Increasing concentration of PRES seems to exert significant effects versus H2O2 group but still significantly reduced as compare to control, but such a statistical comparison is not shown.
  5. Introduction, lane 39: I would change 'few years' with 'last decades'. Citing [Glass et al. Mechanisms underlying inflammation in neurodegeneration. Cell, 2010] would be appropriate.
  6. Data in figure 1 are mean ± SEM? Please specify this in the caption.
  7. Lane 228-229: authors stated "[...], which highlighted a grade 1 signs of cytotoxicity, being about 20% of cells [...]". Please specify the criteria used to discriminate grades of toxicity or remove this sentence. Also check the manuscript for singulars and plurals (e.g. should be 'highlighted a grade 1 sign' or 'highlighted grade 1 signs'.
  8. Please check the manuscript for abbreviations. If you would like to use OS for oxidative stress, spell it the first time and use OS after. Otherwise use oxidative stress w/o abbreviation (this latter option also helps the reader). Same for H2O2 and hydrogen peroxide.
  9. Figure 2 and Figure S1 are the same. Please remove phase-contrast images from figure 2 or remove figure S1. Also scale bars are not visible.
  10. Figure 3: representative pictures of DAPI would be of help.
  11. Figure 5 and figure 6: please avoid to re-scale y-axis that may be misleading visualized (under- or over-estimating effects).

Author Response

  1. Statistical considerations: please insert a section entitled 'Statistical analysis' or 'Statistical consideration' instead of 'Analysis of Data'. Did the authors check data for normality? Which test was used?

We agree with Referee that Statistical analysis is more appropriate than Analysis of data and thus the title of point 2.5 has been changed accordingly. Regarding the check for normality, ANOVA assumes that the data are sampled from populations with identical SDs and that are sampled from populations that follow Gaussian distributions. These criteria were always tested using the method of Bartlett and Kolmogorov-Smirnov, respectively. This has been now stated in point 2.5, Statistical analysis.

  1. For comparisons between group authors used ANOVA followed by Bonferroni post(-hoc) test. This analysis of variance is appropriate for comparisons between n>2 groups and 1 independent variable. In almost all the experiments you have 2 independent variables (H2O2and PRES treatment), which means that multiway ANOVAs is required.

We partially agree with the Referee regarding the use of two-way ANOVA for statistical comparison of the present data. One-way ANOVA compares three or more groups defined by one factor (independent variable). As reported by Harvey Motulsky, [Prism 4 Statistics Guide −Statistical analyses for laboratory and clinical researchers. GraphPad Software Inc., San Diego CA, 2003, pg 68, chapter “Is there only one factor?” or pg 76 (available at https://cdn.graphpad.com/faq/2/file/Prism_v4_Statistics_Guide.pdf)], one way ANOVA analysis is indicated to compare a control group vs a drug treatment group and a group treated with drug plus antagonist (as present study) or to compare a control group vs different concentrations drug treatments, as the “factor” in this case is considered being the “treatment”. The involvement of more than one factor requiring two-way ANOVA is, for example, when you compare three different drugs in men and women: the two factors are in this case the “treatment” and “gender”. As the “treatment” should be considered a single “factor”, this is way we used ONE WAY ANOVA. This applies for all the experiments unless ROS determination, in which a second variable “time” (1h and 24h) was indeed assessed. For these data, we have repeated the analysis by using two-way ANOVA, which however gave superimposable results to previous analysis.

  1. Another issue is related to statistical comparisons between treated groups and control. Eg – Figure 2: Is the cell viability of PRES 0.1 ug/ml group significantly reduced as compared to control (no PRES no H2O2)?
  2. Figure 3, figure 4, figure 5, figure 7 are all example of neglected comparisons. Increasing concentration of PRES seems to exert significant effects versus H2O2group but still significantly reduced as compare to control, but such a statistical comparison is not shown.

We apologize for not having reported the entire statistical comparisons in the Figures: this was made to improve clarity, trying to focus the attention on major results. We agree with the Referee that this might not be appropriate and in the revised version of the manuscript all the comparisons were reported in all Figures.

  1. Introduction, lane 39: I would change 'few years' with 'last decades'. Citing[Glass et al. Mechanisms underlying inflammation in neurodegeneration. Cell, 2010] would be appropriate.

The suggested changes have been performed, as well as the reference of Glass et al quoted. 

  1. Data in figure 1 are mean ± SEM? Please specify this in the caption.

Yes, data in Figure 1 are reported as mean ±SEM. This was already stated in the legend caption.

  1. Lane 228-229: authors stated "[...], which highlighted a grade 1 signs of cytotoxicity, being about 20% of cells [...]". Please specify the criteria used to discriminate grades of toxicity or remove this sentence. Also check the manuscript for singulars and plurals (e.g. should be 'highlighted a grade 1 sign' or 'highlighted grade 1 signs'.

The criteria were already quoted in Mat&Met, point 2.2.4, cell viability assay. These refer to the USP 28 (United States Pharmacopeia edition 2005) grade scale for assessment of the cytotoxic potential of tested materials as follows: grade 0 - no reactivity (discrete intracytoplasmic granules, no cell lysis); grade 1 - slight reactivity (no more than 20% of the cells are round, loosely attached and without intracytoplasmic granules; occasional lysed cells are present); grade 2 - mild reactivity (no more than 50 % of the cells are round and devoid of intracytoplasmic granules, no extensive cell lysis and empty areas between cells); grade 3 - moderate (up to 70% of cells are rounded or lysed); grade 4 - severe (nearly complete destruction of the cells). It was not possible to report this in detail in point 2.2.4 as this text overlaps with our previously published papers, and thus only the appropriate references have been cited. However, as this might be useful to the reader, we have implemented Figure 1 and Figure S1 legends. Singulars and plural were carefully checked. 

  1. Please check the manuscript for abbreviations. If you would like to use OS for oxidative stress, spell it the first time and use OS after. Otherwise use oxidative stress w/o abbreviation (this latter option also helps the reader). Same for H2O2and hydrogen peroxide.

We apologize for the careless in reporting some of the words used. OS instead of oxidative stress is now reported throughout the text (unless the first time), as well as H2O2 instead of hydrogen peroxide (idem).

  1. Figure 2 and Figure S1 are the same. Please remove phase-contrast images from figure 2 or remove figure S1. Also scale bars are not visible.

Figure S1 was inserted as it was requested by Referee 1. We agree, however, that this might constitute a duplicate and thus the panel has been removed from Figure 2 in the manuscript. Scale bars have been also highlighted in both Figure 1 and Figure S1 as requested.

  1. Figure 3: representative pictures of DAPI would be of help.

We agree with Referee that representative pictures of DAPI would be of help. Despite allowing a correct evaluation of the effects elicited by the compounds, however, the photos taken are not of sufficiently high quality as requested by the standards of the Journal, due to technical problems on the microscope camera. If the Referee thinks that these should be shown anyway, we suggest including them in the supplementary info.

  1. Figure 5 and figure 6: please avoid to re-scale y-axis that may be misleading visualized (under- or over-estimating effects). 

Y scale of Figure 5 and 6 has been adjusted as requested

Round 2

Reviewer 1 Report

I apologize for not making my question clear during the first review. What is the scientific rationale to not use a physiological ACSF solution during experimentation? 

This reviewer agrees that a HEPES based ACSF is protective when obtaining slices and improves slice health and helps edema. However, to continue to use this composition of solution during the experiment, ignores the physiological ion composition and increases the buffering capacity solution that could impact the study. 

Author Response

We really apologize with Referee as our previous answer to Q3 was misleading, as we focused on HEPES and not on sodium concentration.

The maintenance of functional living cells in tissue slices is crucial: for this reason, a large number of ACSF formulations can be found in the literature, all prepared with the aim of  keeping them in conditions resembling those found in the intact brain. Regarding sodium concentration, this might vary from 118 mM (when co-present with HEPES without NaHCO3, see the old paper of Delanoy et al published in 1986, PMID 6816242) up to 150 mM (see PMID 3287453), suggesting that there is still no agreement on the “optimal physiological” ACSF composition for healthy maintenance of brain slices.

The main factors affecting the slice viability during the post-slicing period include the stability of its pH (PMID 3287453): this is why we preferred to use a HEPES-ACSF, that is more reliable than ACSF buffered with NaHCO3. That used in the present study, very close to HEPES-Tyrode solution in its composition (see for ex. http://cshprotocols.cshlp.org/content/2007/2/pdb.rec10805.full?text_only=true), contains NaCl 120 mM and HEPES 25 mM. Other components are (in mM): 2 KCl; 1 CaCl2; 1 MgSO4; 1 KH2PO4 and 10 glucose. Final pH was adjusted to 7.3 with NaOH. This composition has been proven to ensure maintenance of healthy brain slices, thus it has been widely used in many labs, including to perform studies to assess neuroprotection (see for ex. PMID 24412540, 29506554, 30387069, 26347375, 23401802), to study drug distribution in the brain (see for ex PMID 19299522, 23336814) or for electrophysiology. This ACSF has been also commonly used in our studies (see for ex PMID 32470081, 27739104, 27923201, 26942555 and others). When we set up the experimental protocol, to check ACSF suitability in our experimental conditions, we measured the time course of LDH and glutamate release from slices over 180 min period: these remained mostly unchanged during time, averaging to 1.54±0.18 µmol/min/mg tis and 85.01±6.55 pmol/mg tiss for LDH and glutamate, respectively (unpublished observation). Moreover, hydrogen peroxide (0-50 mM) and glutamate (0-100 mM) caused a concentration-dependent injury in brain slices in term of MTT assay and slice edema (PMID 27739104), further suggesting a good response of the tissue. Appropriate controls have always been run in each experimental set to avoid any further problem. Finally, we would like to point out that experiments on slices were performed to further validate experiments performed in cells in a tissue context. Giving the present results, which can be considered a “preliminary” assessment of the potential of PRES, we are planning to verify neuroprotective properties of PRES in in vivo models.

Round 3

Reviewer 1 Report

The authors are not giving a scientific rationale as to why the normal ion concentration, that is present in the body, does not apply to their experiments.  

Author Response

We are really sorry that our previous responses to this point were not sufficiently clear. We never state that the “normal” ion concentration does not apply to our experiments. Indeed we tried to explain that within the large number of ACSF formulations for brain slices that can be found in the literature, which contains sodium concentration varying from 118 mM up to 150 mM (PMID 3287453), we selected that which could ensure the best maintenance in pH stability and, at the same time, healthy tissue (i.e. the scientific rationale). We would like to remind that this ACSF was first used with the same purpose by other research groups (see the previously quoted references) and by themselves (idem). Moreover, as stated in our previous response, we performed preliminary tests (LDH and glutamate release, concentration-response dependent effects etc…) to further check the suitability of this ACSF in our experimental conditions.